# Development of a Machine Learning Model to Predict Non-Durable Response to Anti-TNF Therapy in Crohn’s Disease Using Transcriptome Imputed from Genotypes

**DOI:** 10.3390/jpm12060947

**Published:** 2022-06-09

**Authors:** Soo Kyung Park, Yea Bean Kim, Sangsoo Kim, Chil Woo Lee, Chang Hwan Choi, Sang-Bum Kang, Tae Oh Kim, Ki Bae Bang, Jaeyoung Chun, Jae Myung Cha, Jong Pil Im, Min Suk Kim, Kwang Sung Ahn, Seon-Young Kim, Dong Il Park

**Affiliations:** 1Division of Gastroenterology, Department of Internal Medicine and Inflammatory Bowel Disease Center, Kangbuk Samsung Hospital, School of Medicine, Sungkyunkwan University, Seoul 03181, Korea; skparkmd@gmail.com; 2Medical Research Institute, Kangbuk Samsung Hospital, School of Medicine, Sungkyunkwan University, Seoul 03181, Korea; chilwoo.lee@gmail.com; 3Department of Bioinformatics, Soongsil University, Seoul 06978, Korea; yebins96@gmail.com; 4Department of Internal Medicine, College of Medicine, Chung-Ang University, Seoul 06973, Korea; gicch@cau.ac.kr; 5Department of Internal Medicine, Daejeon St. Mary’s Hospital, College of Medicine, Catholic University, Daejeon 34943, Korea; sangucsd@gmail.com; 6Department of Internal Medicine, Haeundae Paik Hospital, College of Medicine, Inje University, Busan 48108, Korea; kto0440@paik.ac.kr; 7Department of Internal Medicine, College of Medicine, Dankook University, Cheonan 31116, Korea; kibaebang@gmail.com; 8Department of Internal Medicine, Gangnam Severance Hospital, College of Medicine, Yonsei University, Seoul 06273, Korea; j40479@gmail.com; 9Department of Internal Medicine, Kyung Hee University Hospital at Gang Dong, College of Medicine, Kyung Hee University, Seoul 05278, Korea; clicknox@hanmail.net; 10Department of Internal Medicine and Liver Research Institute, College of Medicine, Seoul National University, Seoul 03080, Korea; jpim0911@snu.ac.kr; 11Department of Human Intelligence and Robot Engineering, Sangmyung University, Cheonan 31066, Korea; minsuk.kim@smu.ac.kr; 12Functional Genome Institute, PDXen Biosystems Inc., Suwon 16488, Korea; kwangsung.ahn@gmail.com; 13Personalized Medicine Research Center, Korea Research Institute of Bioscience and Biotechnology (KRIBB), Daejeon 34141, Korea; kimsy@kribb.re.kr

**Keywords:** genotype, genetic features, anti-TNF, Crohn’s disease

## Abstract

Almost half of patients show no primary or secondary response to monoclonal anti-tumor necrosis factor α (anti-TNF) antibody treatment for inflammatory bowel disease (IBD). Thus, the exact mechanisms of a non-durable response (NDR) remain inadequately defined. We used our genome-wide genotype data to impute expression values as features in training machine learning models to predict a NDR. Blood samples from various IBD cohorts were used for genotyping with the Korea Biobank Array. A total of 234 patients with Crohn’s disease (CD) who received their first anti-TNF therapy were enrolled. The expression profiles of 6294 genes in whole-blood tissue imputed from the genotype data were combined with clinical parameters to train a logistic model to predict the NDR. The top two and three most significant features were genetic features (*DPY19L3*, *GSTT1*, and *NUCB1*), not clinical features. The logistic regression of the NDR vs. DR status in our cohort by the imputed expression levels showed that the β coefficients were positive for *DPY19L3* and *GSTT1*, and negative for *NUCB1*, concordant with the known eQTL information. Machine learning models using imputed gene expression features effectively predicted NDR to anti-TNF agents in patients with CD.

## 1. Introduction

Crohn’s disease (CD) is a chronic relapsing inflammatory bowel disease (IBD) that causes progressive bowel damage and disability [1]. Monoclonal anti-tumor necrosis factor α (anti-TNF) antibodies have revolutionized the care of patients with CD, enabling the achievement of clinical and endoscopic remission [2]. However, despite the established efficacy of these drugs, one-fifth of patients will not respond to these agents (primary non-response (PNR)), and an additional one-third will eventually fail therapy (secondary loss of response, non-durable response (NDR)), requiring an additional or changed medication or surgery [3,4,5]. The exact mechanisms of PNR and NDR remain poorly defined. There is currently considerable interest and an unmet need for the use of genetic markers to predict therapeutic responses. Most prior studies examining this question studied one or a few candidate genes, had small sample sizes, and did not yield definitive results [6,7,8,9,10,11,12,13,14]. Polymorphisms in TNF-α [8], IBD5 locus [15], immunoglobulin G (IgG) Fc receptor IIIa [16], autophagy (ATG16L1) [17], and apoptosis-related genes [6] have also been variably associated with a response to anti-TNF agents. A limitation of exclusively studying a few candidate loci or IBD-risk alleles is the possibility of missing potentially relevant associations across loci that more broadly influence immune function across a disease spectrum.

It is tempting to develop machine learning models for the prediction of the durable response (DR) status of anti-TNF therapy based on patient genotypes, as they are non-invasive and baseline in nature. However, it is challenging because there are numerous patient genotype markers to consider, and the number of cases is limited. As far as we know, there have been no reports on genotype-based markers that can predict future NDR in CD. The straightforward use of genome-wide genotype data for model training is usually hampered by overfitting (data not shown). Hence, here we propose a novel approach to dimensionality reduction by transforming the genotype dataset into a gene expression dataset (Figure 1).

A transcriptome-wide association study that tests the association between the phenotype and the “imputed” gene expression has been successfully applied in many cases. The PrediXcan is one such method that imputes gene expression from genotypes using machine learning models developed based on the GTEx genotype and the corresponding transcriptome datasets of various tissues. In this study, we applied the PrediXcan to our genome-wide genotype data to impute gene expression and used the resulting expression values as well as clinical parameters as features for training machine learning models for the prediction of NDR vs. DR status. Thus, the number of features was reduced from millions to thousands. Notably, imputed expression values are not direct experimental observations; rather, they are derived features upon the combination of experimental values in a mathematically defined manner.

## 2. Materials and Methods

### 2.1. Study Population

A total of 894 IBD patients were recruited from the Identification of the Mechanism of the occurrence and Progression of Crohn’s disease through integrated Analysis on both genetiC and environmenTal factors (IMPACT), UC multiomics, and Occurrence of Anti-drug antibody and change of drug level after CT-P13 therapy and their Impact on clinical outcomes in moderate to Severe inflammatory bowel disease (OACIS) cohorts. IMPACT was a prospective multicenter study established in Korea in 2017. Clinical data and biological specimens (including blood, stool, and tissue specimens) of CD patients who were newly diagnosed or followed up within 16 university hospitals were collected. The ulcerative colitis (UC) multiomics study was a prospective multicenter study, established in Korea in 2020. A total of 14 university hospitals participated in this study and collected clinical data and biological specimens (including blood, stool, tissue, and saliva specimens) from UC patients. Details of the IMPACT and UC multiomics study cohorts have been previously described [18]. The OACIS study was a prospective multicenter observational study conducted at 18 university hospitals in Korea between August 2016 and September 2019. Consecutive patients older than 18 years with moderate to severe active CD or UC who started CT-P13 therapy were prospectively enrolled in the study.

Among the 894 patients, the inclusion criteria were as follows: (1) diagnosis of CD and (2) first anti-TNF therapy consisting of infliximab or CT-P13. Using a chart review, two study investigators (S.K.P., D.I.P.) characterized the patients’ responses to their first anti-TNF therapy. All patients received standard induction dosing (infliximab or CT-P13 5 mg/kg at weeks 0, 2, and 6, and every 8 weeks thereafter). DR was defined as the maintenance of the response to anti-TNF therapy for at least 24 months after initiation. NDR was defined as non-response within 24 months after starting therapy accompanied by an alteration in therapy (addition or escalation of corticosteroids, switch to a different agent, or surgery). Patients with primary non-response (non-response at 12 weeks after starting therapy), those who ceased treatment due to adverse effects before the 24-month time point, and those who experienced adverse events related to a loss of response (for example, infusion reactions due to immunogenicity) were classified as NDR.

### 2.2. Genotyping

Blood samples from the three cohorts were used for genotyping with the Korea Biobank Array, which comprises 833,535 single-nucleotide proteins (SNPs) [19]. Sample quality control of call rate (>95%), heterozygosity (within ±3 standard deviation of mean), relative relationship (proportion of IBD < 0.2), and principal component analysis (within the main cluster) tests eliminated 16 samples, resulting in 878 samples. All QC metrics were calculated with PLINK (v1.90b6.4). The genotype data of 749,383 SNPs that survived the quality control analysis were imputed with the Korean reference panel using the BEAGLE software package [20]. The reference panel comprised 28,445 samples that were genotyped with the Korea Biobank Array and subsequently imputed with the East Asian population of the 1000 Genomes data. After removal of the SNPs with minor allele frequency < 0.05 or that violated the Hardy–Weinberg equilibrium (*p* < 0.001), a total of 6,153,437 SNPs were available for analysis.

Among the 878 samples that passed filtering, 234 patients met the inclusion criteria (220 and 14 with DR and NDR, respectively). Table 1 shows the patients’ demographic and epidemiological characteristics. For expression quantitative trait locus (eQTL) analysis of the association signals, GTEx [21] and the CD eQTL database from the Asan Medical Center IBD eQTL Browser (http://asan.crohneqtl.com/ accessed on 30 October 2021) were utilized [22].

### 2.3. Imputing Gene Expression from Genotype

Machine learning models for imputing tissue-specific gene expression from genotypes were downloaded from the PrediXcan website (https://predictdb.org/post/2017/11/29/gtex-v7-expression-models/ accessed on 21 April 2021). Each was a linear regression model that was trained on the European subset of the genotype and expression datasets compiled by GTEx v7. The PrediXcan algorithm considers only SNPs proximal to the target gene and further limits them by employing a feature selection method based on an elastic net. The imputed expression value of the target gene is then obtained by taking the scalar product of the regression coefficients with the alternative allele counts of the surviving SNPs. It should be noted that the imputability of a gene varies across tissues, and PrediX can evaluate imputability through cross-validation, releasing only the models passing preset filtering criteria. Among the tissue models that PrediXcan v7 released, we considered that gene models for the whole blood, transverse colon, and small intestine terminal ileum were relevant to CD, focusing the subsequent analyses on these tissues only. We calculated the imputed expression values of 6294, 5612, and 3107 genes for 234 patients in those 3 tissues, respectively. The imputed expression values were used as features in the predictive model development, as described below. The selected genes were also tested for their associations with NDR/DR status using the univariate logistic regression implemented as a PrediXcan function.

### 2.4. Development of Predictive Machine Learning Models

#### 2.4.1. Preprocessing

The following clinical variables—age, sex, smoking status, family history, disease location at diagnosis (including upper gastrointestinal involvement), disease behavior at diagnosis (including perianal involvement), and gene expression values imputed by the PrediXcan—were used as the features for the predictive model for discriminating between NDR (cases) and DR (controls). Several functions in scikit-learn [23] have been used to encode and standardize variables. For example, numerical variables, such as age and expression values, were standardized using StanardScaler. Dichotomous categorical variables were encoded as 0 or 1 using LabelEncoder, whereas multiclass features were binarized using OneHotEncoder.

#### 2.4.2. Model Training and Feature Selection

A total of 234 samples were split into training and test sets in an 8:2 ratio using the train_test_split of scikit-learn, while constraining the 14 cases to be split into 10 and 4 in the training and test sets, respectively. For the model training, we used logistic regression of scikit-learn, as it is simple and sufficiently effective for large cycles of iterative training with an additional option of feature selection, such as Least Absolute Shrinkage and Selection Operator (LASSO) and elastic net. Our feature selection process proceeded in two stages. In the first stage, we leveraged the LASSO penalty in the logistic regression training to reduce the number of features. Specifically, the C parameter in logistic regression was scanned from 10 to 100 in increments of 10, and the trained model was evaluated by 5-fold cross-validation of the area under the receiver operating characteristic curve (AUC-ROC). Using the liblinear optimizer in the logistic regression, 10–1000 features survived. The overall scheme of the model training is shown in Figure 2.

In the second stage of the feature selection process, we used recursive feature elimination (RFE) of scikit-learn, which iterates model training by removing the least important feature in each round until the desired number of features survive. Using the selected features, the model performance was measured using the test set. We repeated the entire feature selection stage 100 times with random shuffling of the training and test split. The most frequent feature combinations were sought from this repetition.

As our data are somewhat imbalanced and limited in size, there is a concern that AUC-ROC can discern different models. For each model, we also evaluated the precision-recall curve and AUC-PRC. Given an ROC curve, the Youden’s index, sensitivity + specificity, was calculated, and the cutoff was defined where the Youden’s index was maximal. With this cutoff, sensitivity (recall), specificity, and precision of the model were assessed. The mean and standard deviation of these metrics from the 100 repetitions were then reported. Among the 100 repetitions, the case whose metrics were the most similar to the mean values was chosen and its ROC and PRC curves were presented as representative.

## 3. Results

### 3.1. DR/NDR Prediction Models Based on Different Sampling Tissues

GTEx collected the genotype data and corresponding expression datasets from various sampling tissues. Accordingly, the PrediXcan provides a gene expression imputation model, one for each gene in each sampled tissue. The number of imputable genes varies across tissues. As we were interested in predicting the NDR to infliximab in IBD patients, we downloaded the imputable gene models on three sampling tissues: whole blood (6294 genes), colon transverse (5612 genes), and small intestine terminal ileum (3107 genes). First, we imputed the gene expression values from our genotype dataset by using these models for each tissue. Second, the imputed expression values were used as features in training NDR/DR response models based on logistic regression with the feature selection scheme based on the LASSO penalty and RFE. The performance of the model at each selection step was evaluated using 5-fold cross-validation of the AUC-ROC. The test performance was measured for the best model. Since our dataset was limited to a small number of NDR samples, our results may not be stable and are likely to suffer from overfitting. To overcome this limitation, we repeated the entire feature selection process 100 times and searched for a recurring combination of features.

As the first step of the proof of concept, we selected only one feature from the feature selection processes involving the LASSO and RFE. Figure 3 shows the distribution of the predictive performance of these models during 100 repetitions in three different sampling tissues. Although the performance with the test set varied approximately twice as much as that with the training set, the medians were very similar to each other, implying that the extent of overfitting was negligible. See Appendix A for the distributions of AUC-PRC and Appendix A for representative ROC and PRC curves.

Among the three different tissue models, the imputed expression in whole blood achieved the highest performance, and the difference in performance among the models from different tissues implies that genetic factors contribute significantly to the predictability of NDR/DR status. Indeed, the features that were selected most frequently were genes in all three tissue models (Table 2 and Appendix A).

Interestingly, Dpy-19-like C-mannosyltransferase 3 (*DPY19L3*) was selected highly consistently (79 of 100 trials) from the random shuffling of the training/test dataset split in the whole-blood model, which performed the best. On the other hand, two different genes, *TXNDC16* and *ENSG00000270127*, were most frequently selected in the other two tissue models. Furthermore, their selections were less consistent than those of the whole-blood model. Based on these observations, we focused on the whole-blood model in the following analyses.

### 3.2. Selection of Top Two and Three Features Using Whole-Blood Model

We selected the top two and three most significant features by combining the LASSO penalty and the RFE. These selections were independently performed for each of the 100 random shufflings. As shown in Table 3, only the genetic features were selected. *DPY19L3*, the single most frequently selected feature, was included in the most frequent combinations of two and three features. Glutathione S-transferase theta 1 (*GSTT1*) was the most frequent two- and three-feature selection feature.

As shown in Appendix A, more features were included and better performance was achieved, although the difference between the two- and three-feature models was not large.

### 3.3. Contribution of Clinical Features

Although eight clinical features were included in the training, none survived the feature selection process when we selected up to three features. To evaluate the contribution of clinical features in predicting NDR/DR status, we built a logistic regression model based solely on them and compared it with the models built by infusing the genetic features selected in the previous section. Using the same scheme of random shuffling of the training/test split, the performances were evaluated (Appendix A). The model with only the clinical features did not perform as well as the single gene model in terms of AUC-ROC (*DPY19L3*; 0.568 vs. 0.845 for the training set, 0.603 vs. 0.839 for the test set). In contrast, infusing genes into the model with only the clinical features dramatically improved its performance (Figure 4). This trend is similar to that observed in gene-only models. Although the inclusion of clinical features did not show clearly improved performance over the gene-only models in terms of AUC-ROC or AUC-PRC (Appendix A), the average sensitivity, specificity, and precision were improved with the inclusion of the clinical features (Appendix A).

### 3.4. Genetic Bases of Selected Features

PrediXcan develops models for the imputation of gene expression based on the genotypes and transcriptomics datasets compiled by GTEx [24], and as most of the whole-blood samples collected by GTEx were of European origin [25], there might be concern about whether the gene expression models from PrediXcan are valid for the current study population of Korean origin. To address this, we examined the genetic basis of the three genes identified in this study. The association of the imputed expression values with NDR/DR status was analyzed by univariate logistic regression for each gene using the PrediXcan function. As shown in Table 4, all showed significant associations (*p* < 0.01). The logistic regression coefficients (β) of *DPY19L3* and *GSTT1* were positive, meaning that their higher imputed expression levels increase the probability of being NDR, while that of *NUCB1* was negative, decreasing the probability of being NDR.

For *DPY19L3*, many eQTLs are known in whole blood according to the GTEx; for example, the alternative allele of rs4805759 has a net effect size of −0.23 (*p* = 5.2 × 10^−38^). The eQTL database constructed for the whole-blood samples of Korean CD patients [22] also shows the same direction of effect, that is, this alternative allele has the regression slope of −0.478 (*p* = 9.9 × 10^−6^). In our genotype data, the alternative allele frequencies were 0.8341 and 0.4286 for DR and NDR samples, respectively (*p* = 2.2 × 10^−5^). The finding that the NDR samples have fewer alternative alleles of rs4805759 than DR and its eQTL regression coefficient is negative is in agreement with the univariate logistic regression coefficient (β) of *DPY19L3* being positive (Table 4). For *GSTT1*, the Korean eQTL database lists three SNPs as eQTLs (rs368588, rs2236620, and rs2236621, with slopes of 0.683–1.186), whereas no eQTLs were identified in GTEx. While *NUCB1* is not listed in the Korean eQTL database, its eQTLs are listed in the GTEx (max. net effect size of −0.45 for rs10415881). Their directions of effect were also in agreement with the respective univariate logistic coefficients (β).

## 4. Discussion

A recent trend in large-scale association studies is transcriptome-wide association studies that transform genome-wide genotype datasets into imputed gene expression datasets to identify gene–trait associations. The PrediXcan is one such method that imputes tissue-specific gene expression from genotypes by the machine learning models that have been pre-developed based on GTEx datasets. In this study, we applied the PrediXcan to our genome-wide genotype data and used the resulting expression values in whole-blood tissue as well as clinical parameters as features in training machine learning models for predicting NDR vs. DR status. The selected top two and three most significant features were genetic features only (*DPY19L3*, *GSTT1*, and *NUCB1*). Adding clinical features without further feature selection showed slight improvement in the performance over the gene-only models. The logistic regression of the NDR vs. DR status by the imputed expression levels of *DPY19L3, GSTT1,* and *NUCB1* were consistent with the known eQTL information retrieved from GTEx and the Korean CD eQTL database [19]. It was reported that these eQTL databases had concordant directions of eQTL for more than 96% of the target genes [19]. This supports the validity of the PrediXcan models based on the European data applied to the Korean cases. 

Among the genetic features, *DPY19L3* was the most frequently selected. *DPY19L3*-mediated C-mannosylation of R-spondin1 at W^156^ is required for R-spondin1 secretion [26]. R-spondin1 is a secreted protein that enhances Wnt signaling, an important pathway for immune cell maintenance and renewal [26]. Wnt signaling in immune cells is very diverse, for example, the tolerogenic role of dendritic cells, development of natural killer cells, thymopoiesis of T cells, B-cell-driven initiation of T cells, and macrophage actions in tissue repair, regeneration, and fibrosis [27]. As the imputed expression level of *DPY19L3* is supposed to be higher in NDR than in DR, the inflammatory response of immune cells might not be controlled by an anti-TNF agent.

*GSTT1* was the most frequently used two-feature selection method. Among the members of GSTs, glutathione S-transferase theta 1 (*GSTT1*) and GSTM1 in particular have become recent targets of active investigation into their role in increased susceptibility to IBD, Behçet’s disease, or other autoimmune diseases such as primary sclerosing cholangitis. *GSTT1* contributes to detoxifying chemicals, including reactive oxygen species (ROS) [28]. In a previous study using a DSS-induced colitis mouse model [28], the authors noted attenuation of colitis through gene transfer of *Gstt1* via an IL-22-dependent pathway. Downregulation of *GSTT1* by the pathogen-associated molecular patterns of microbes reduces innate defense responses and goblet cell differentiation. *GSTT1* ameliorates colitis, and its mutations are linked to chronic intestinal inflammation due to insufficient dimerization.

Impaired ROS production due to inactivation of patient variants in genes encoding nicotinamide adenine dinucleotide phosphate oxidases as ROS sources is associated with CD and pancolitis, whereas ROS overproduction due to upregulation of oxidases or altered mitochondrial function has been linked to ileitis and ulcerative colitis [29]. The major role of TNF is to regulate the immune system through the activation of TNF receptors and downstream pathways involving molecules, such as nuclear factor kappa-B, mitogen-activated protein kinases, caspases, and ROS/reactive nitrogen species [30]. As the imputed expression level of *DPY19L3* is supposed to be higher in NDR than in DR, it is hypothesized that ROS levels uncontrolled by anti-TNF agents may be related to a continuous inflammatory response. 

*NUCB1*, which is involved in three-feature selection, interacts with *GNAI1* or *GNAI3* to activate them [31]. In a mouse model, *GNAI1* and *GNAI 3* suppressed DSS-plus-azoxymethane-induced colon tumor development, whereas the expression of *GNAI2* in CD11c^+^ cells and interleukin-6 (IL6) in CD4^+^/CD11b^+^ dendritic cells appeared to promote these effects. As GNAI1 and GNAI3 block IL6 signaling to inhibit inflammation or tumorigenesis, strategies to induce GNAI1 or block GNAI2 and IL6 have been suggested to prevent or treat colitis-associated cancer [32]. As the imputed expression level of *NUCB1* is supposed to be lower in patients with NDR, the actions of GNAI1 or GNAI3 and blocking IL6 signaling may be impaired. Despite anti-TNF agent use, the inflammatory response by IL6 might result in NDR.

The strength of our study is that we found genetic features using a machine learning method with genome-wide genotype data and the resulting expression values. Although we used the PrediXcan model based on the European origin genotypes and transcriptomic datasets compiled by GTEx, it was valid for the current Korean population. Since the gene expression value used as a feature is imputed by an individual’s unique genotype, it can accordingly be viewed as an expected baseline gene expression value, rather than being related to disease onset or drug administration.

There are also some limitations to this study. First, our definitions of NDR were based on clinical evidence from chart reviews rather than from prospectively collected data. Future studies should prospectively include comprehensive clinical, endoscopic, and radiological evidence to define the response. Second, imputed expression values are not direct experimental observations; rather, they are derived features by the combination of experimental values in a mathematically defined manner. Third, except for eight clinical variables, drug history, such as combinations of immunosuppressants, was not considered in this analysis. Including more clinical variables and comparing with genetic features in the predictive model are needed in future studies.

## 5. Conclusions

In conclusion, machine learning models with transcriptomes imputed from genome-wide genotype datasets effectively predicted NDR to anti-TNF agents in patients with CD. However, the genetic features derived from our study require validation in another cohort, whereas the pathway of genetic features associated with NDR to anti-TNF agents requires further study. 

## Figures and Tables

**Figure 1 jpm-12-00947-f001:**
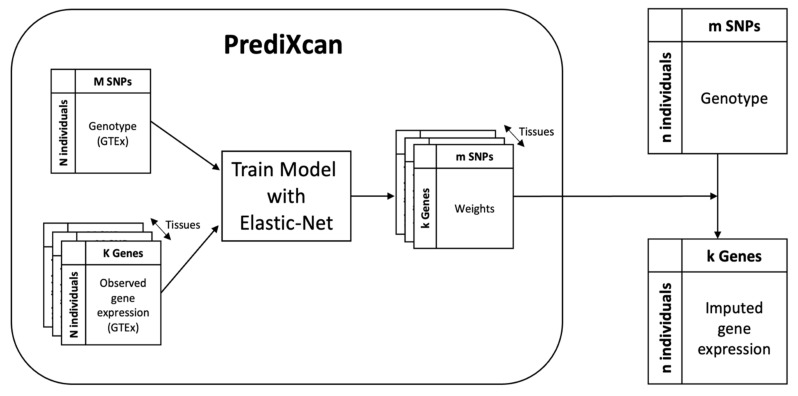
Overview of imputing gene expression from genotype. The PrediXcan models (round-cornered box) were downloaded from https://predictdb.org (accessed on 21 April 2021). They were developed for a number of tissues using the matched genotype and gene expression datasets compiled by the GTEx consortium. A given gene’s expression value was linearly modeled from the genotypes of neighboring single-nucleotide proteins (SNPs), which was selected through elastic net. By applying the linear regression weights to our genotype data (upper right), we were able to impute the gene expression in our tissue of interest (lower right).

**Figure 2 jpm-12-00947-f002:**
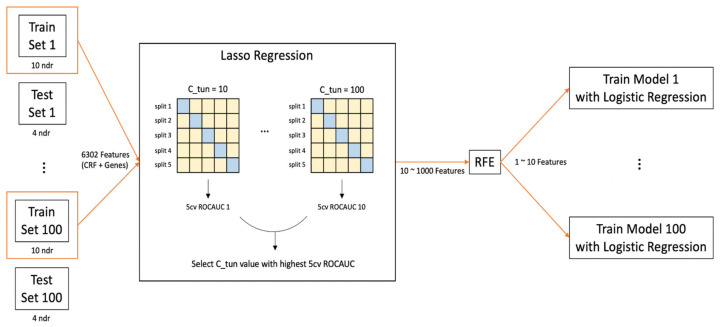
The overall model training scheme. The dataset was split into training and test sets in an 8:2 ratio. This random split was repeated 100 times. For each split, the model training involving Least Absolute Shrinkage and Selection Operator (LASSO) regression, and recursive feature elimination (RFE) was performed. In the LASSO regression, the C parameter was scanned from 10 to 100 in multiples of 10 for the highest 5-fold cross-validation (5-CV) area under the receiver operating characteristic curve (AUC-ROC) value. For the best C parameter, typically 10 to 1000 features survived. Among these features, a fixed number, ranging from 1 to 10, of features was selected through RFE. For a given number of the selected features, 100 different models were developed due to the 100 random data splits. The model performance was evaluated with the test set using logistic regression of the selected feature(s). For 5-CV, the training and test sets are shown in light yellow and blue, respectively.

**Figure 3 jpm-12-00947-f003:**
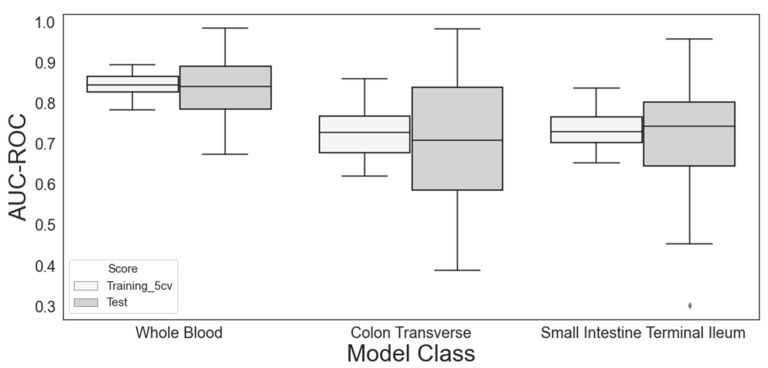
Performance of the three tissue models used for gene expression imputation in classifying the non-durable response (NDR) vs. the durable response (DR). The model training by 5-fold cross-validation and feature selection via LASSO and recursive feature elimination was repeated 100 times (see main text for details). The single most significant gene was selected from each trial. The training and test performances given as area under the receiver operating characteristic curve (AUC-ROC) are shown as boxplots. That from the whole-blood model was significantly higher than that from the colon transverse (*p_t_*_-test_ = 6.5 × 10^−38^ and 3.7 × 10^−12^ for the training and test sets, respectively) or the small intestine terminal ileum (*p_t_*_-test_ = 1.8 × 10^−46^ and 7.5 × 10^−15^ for the training and test sets, respectively).

**Figure 4 jpm-12-00947-f004:**
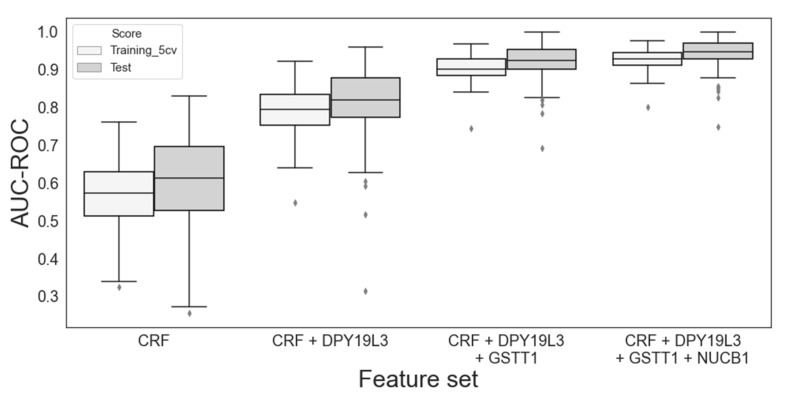
Performance evaluation of the models built with varying sets of features for classifying the non-durable response (NDR) vs. the durable response (DR). The feature genes were the same as listed in Table 3. A total of eight patient clinical parameters (see Methods Section), denoted as *CRF* in the figure, were also included in the model without further feature selections.

**Table 1 jpm-12-00947-t001:** Patients’ clinical characteristics.

	Non-Durable Response (n = 14)	Durable Response (n = 220)	*p*-Value
Age at diagnosis, year (SD)	26.3 (9.2)	28.2 (9.1)	0.45
Gender, male (%)	8 (57.1%)	162 (73.6%)	0.21
History of smoking, n (%)	5 (35.7%)	47 (21.4%)	0.20
Family history of IBD, n (%)	0 (0%)	7 (3.2%)	1.0
Disease duration, year (SD)	9.1 (5.5)	7.5 (3.8)	0.16
Disease location, n (%)			0.07
Ileal	7 (50%)	52 (23.6%)	
Colonic	2 (14.3%)	31 (14.1%)	
Ileocolonic	5 (35.7%)	137 (62.3%)	
Upper GI involvement, n (%)	0 (0%)	11 (5.0%)	0.39
Disease behavior, n (%)			0.35
Inflammatory	9 (64.3%)	163 (74.1%)	
Stricturing	1 (7.1%)	25 (11.4%)	
Penetrating	4 (28.6%)	32 (14.5%)	
Perianal disease, n (%)	6 (42.9%)	85 (38.6%)	0.75
Combination immunosuppressants, n (%)	10 (71.4%)	206 (93.6%)	<0.001
Intestinal resection, n (%)	6 (42.9%)	61 (27.7%)	0.22

IBD, inflammatory bowel disease; GI, gastrointestinal.

**Table 2 jpm-12-00947-t002:** Most frequently selected single feature for each tissue expression imputation model and its performance in classifying NDR vs. DR.

Tissue Expression Model	Selected Feature	Selection Frequency	AUC-ROC (SD)	
			Training 5-CV Set	Test Set
Whole blood	*DPY19L3*	79/100	0.845 (0.027)	0.839 (0.070)
Colon transverse	*TXNDC16*	40/100	0.728 (0.060)	0.711 (0.150)
Small intestine terminal ileum	*ENSG00000270127*	14/100	0.738 (0.050)	0.720 (0.120)

5CV, five-fold cross-validation; AUC-ROC, area under the receiver operating characteristic curve; SD, standard deviation; NDR, non-durable response; DR, durable response.

**Table 3 jpm-12-00947-t003:** The most frequently selected combination of two or three genes for the whole-blood expression imputation model and its performance for classifying NDR vs. DR.

No. of Features	Selected Feature Set	Selection Frequency	AUC-ROC (SD)	
			Training 5CV Set	Test Set
1	*DPY19L3*	79	0.845 (0.027)	0.839 (0.070)
2	*DPY19L3, GSTT1*	32	0.918 (0.023)	0.919 (0.040)
3	*DPY19L3, GSTT1, NUCB1*	9	0.935 (0.024)	0.935 (0.040)

5CV, five-fold cross-validation; AUC-ROC, area under the receiver operating characteristic curve; SD, standard deviation; NDR, non-durable response; DR, durable response.

**Table 4 jpm-12-00947-t004:** Univariate logistic regression analysis of the association between gene expression and NDR/DR status.

Gene Name	Chr	*p*-Value	β Value
*DPY19L3*	19	0.000965	2.703
*GSTT1*	22	0.00343	1.735
*NUCB1*	19	0.00684	−2.142

Chr, chromosome; NDR, non-durable response; DR, durable response.

## Data Availability

The genotype data generated and analyzed in the current study are not publicly available due to the limitation of study consent regarding repository deposition, but they are available from the corresponding author upon reasonable request. The imputed gene expression datasets with the clinical phenotype information are available from the zenodo database (https://doi.org/10.5281/zenodo.6464129, accessed on 16 April 2022).

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
