# Peer review of "Development of a Machine Learning Model to Predict Non-Durable Response to Anti-TNF Therapy in Crohn’s Disease Using Transcriptome Imputed from Genotypes"

_jpm, 2022, doi:10.3390/jpm12060947_

Round 1

Reviewer 1 Report

In this study, the author built a model using imputed gene expression as features that can effectively predict NDR to anti-TNF agents in patients with CD.

I have a few concerns:

  1. In abstract, I do not understand “The logistic regression coefficients (β) in our cohort showed that the predicted expression levels of DPY19L3 and GSTT1 were higher in the NDR than in the durable response (DR), whereas that of NUCB1 was negative, corresponding to under expression of NDR over DR”, why does the positive β mean a high expression levels and the negative expression β means a under expression? In my expression the β value only show the importance/power of the predictors(features), pleas correct me. And what does “predicted expression levels”, the gene levels were imputed from genotypes and used as the features (inputs), why here you say “predicted expression levels”?
  2. In abstract, the last sentence “Machine learning models using imputed gene expression features effectively predicted NDR to anti-TNF agents based on genetic features in patients with CD.” The gene expressions were used as the features, why you say based on “genetic features” ? Please refine it, do not confusing the readers.
  3. Why do not use the gene expression data directly (not the imputed expression data), or use the genotype data directly to build the model? Please show the performance of the model built directly on the expression data or directly on the genotype data?
  4. Texts in the figure3,4 are two small.
  5. The samples are imp lances, the AUC-ROC can not really tell the performance of the model, please provide the precision, recall value and the area under precision-recall curves.

Author Response

<Reviewer 1>

  1. In abstract, I do not understand “The logistic regression coefficients (β) in our cohort showed that the predicted expression levels of DPY19L3 and GSTT1 were higher in the NDR than in the durable response (DR), whereas that of NUCB1 was negative, corresponding to under expression of NDR over DR”, why does the positive β mean a high expression levels and the negative expression β means a under expression? In my expression the β value only show the importance/power of the predictors(features), pleas correct me. And what does “predicted expression levels”, the gene levels were imputed from genotypes and used as the features (inputs), why here you say “predicted expression levels”?

Response:

Thank you for your important comments. In our logistic regression, the outcome variable was coded as 1 for NDR and 0 for DR. If the explanatory variable, the predicted expression, is associated with the outcome, the corresponding regression coefficient, β, would be non-negligible. Its statistical significance is typically tested by the z-value which is defined as β/se(β), where the denominator is the standard error. The absolute value of β is related to the importance of the predictor, while its sign is related to the direction of effect. If it is positive, higher value of the predictor increases the probability of being NDR, vice versa. Our logistic regression tests the association between the imputed expression level and NDR/DR status, not the differential expression. As pointed out by the reviewer, the terminology, “predicted expression”, is misleading. Hence we have revised it as “imputed expression” throughout the manuscript, and revised sentence in the abstract as follows : “The logistic regression of the NDR vs DR status in our cohort by the imputed expression levels showed that the β coefficients  were positive for DPY19L3 and GSTT1, and negative for NUCB1, concordant with the known eQTL information.”

2.In abstract, the last sentence “Machine learning models using imputed gene expression features effectively predicted NDR to anti-TNF agents based on genetic features in patients with CD.” The gene expressions were used as the features, why you say based on “genetic features” ? Please refine it, do not confusing the readers.

Response: Thank you for your important comments. We revised the sentence to “Machine learning models using imputed gene expression features effectively predicted NDR to anti-TNF agents in patients with CD”.

3.Why do not use the gene expression data directly (not the imputed expression data), or use the genotype data directly to build the model? Please show the performance of the model built directly on the expression data or directly on the genotype data?

Response: There are interests in discovering baseline blood biomarkers of future NDR to anti-TNF therapy in CD. As such, Gaujoux et al (Gut 2018) reported that whole blood expression of TREM-1 was downregulated in NDR at baseline, with a predictive response (AUC 0.94). On the other hand, Verstockt et al (Gut 2018 & eBioMedicine 2019) reported upregulation of baseline whole blood TREM-1 in NDR with a predictive response (AUC 0.80). However, these studies attempted to validate blood expression of the predictor genes identified from gene expression analyses of tissue biopsy samples. As far as we know, there have been no genome-wide whole blood gene expression studies of NDR/DR patients and predictive modeling thereof. We do not have such data, either.

Genotype-based predictive models would be useful, as they are non-invasive and baseline in nature. As far as we know, there have been no reports on genotype-based markers that can predict NDR vs DR in CD. As mentioned in Introduction, we have tried a direct use of genotype data without success. A typical overfitting problem was encountered: a perfect score for the training set but a random prediction for the test set. As such, we did not show the data for this. Unlike the direct use of genotype data, the imputed expression,  meta-features derived from the genotypes, were manageable in our predictive modeling.  Thus we added references (Ref 12,13,14) in the INTRODUCTION of revised manuscript and revised sentence as follows: “It is tempting to develop machine learning models for the prediction of durable response (DR) status of anti-TNF therapy based on patient genotypes, as they are non-invasive and baseline in nature. However, it is challenging because there are numerous patient genotype markers to consider and the number of cases is limited. As far as we know, there have been no reports on genotype-based markers that can predict future NDR in CD. The straightforward use of genome-wide genotype data for model training is usually hampered by overfitting (data not shown). Hence, here we propose a novel approach to dimensionality reduction by transforming the genotype dataset into a gene expression dataset (Figure 1)“

4.Texts in the figure3,4 are two small.

Response: Yes, we revised the figures in the revised manuscript.

5.The samples are imp lances, the AUC-ROC can not really tell the performance of the model, please provide the precision, recall value and the area under precision-recall curves.

Response: We understand the reviewer’s concern on the imbalanced data. While we used AUC-ROC as the metric for training and testing the models, the accompanying AUC-PRC was also measured as suggested. As the test set was small in size, we repeated train-test splits 100 times, examining the collective trends. The boxplots in Figures 3 and 4 show the distribution of AUC-ROC values during the random shuffling. The accompanying AUC-PRC distributions show similar trends as those of AUC-ROC distributions, increasing with more genes included (Supplementary Figures S1, S3, and S5). For each repetition, we defined a cutoff where the Youden’s index was maximal, and measured the sensitivity (recall), specificity, and precision for the test set. The mean and standard deviation of these metrics are shown in Supplementary Tables S4. While the mean sensitivity (recall) was maximal with two genes and dropped slightly with three genes, the mean specificity and precision were improved with more genes.  In order to show a representative ROC and PRC curves, we looked for a case from the 100 repetitions whose metrics are the most similar to these mean values. These curves are shown in Supplementary Figures S2, S4, and S6. The PRC curves were well above the random prediction (the proportion of cases in the test samples). As the ROC curves show better performance with more genes, the PRC curves also showed similar trends. Since the best mean precision in our models was 0.488, there must have been substantial false positives in our predictions. If the primary goal of the clinical predictions such as ours were to screen potential NDR patients that should be closely followed up, we believed that these levels of false positives were acceptable. We added sentence in the METHOD of the revised manuscript as follows: “As our data is somewhat imbalanced and limited in size, there is a concern that AUC-ROC can discern different models. For each model, we also evaluated precision-recall curve and AUC-PRC. Given an ROC curve, the Youden’s index, sensitivity + specificity, was calculated and the cutoff was defined where the Youden’s index was maximal. With this cutoff, sensitivity (recall), specificity, precision of the model were assessed. The mean and standard deviation of these metrics from the 100 repetitions were then reported. Among the 100 repetitions, the case whose metrics were the most similar to the mean values was chosen and its ROC and PRC curves were represented as a representative.” And added supplementary figure S2-S6 and supplementary table S1-S4 accordingly.

Reviewer 2 Report

In the current study, the authors investigated whether durable response (DR) to anti-TNF can be predicted from non-durable response (NDR) using gene expression as imputed from the genetics. Overall, the concept is novel and interesting and the clinical need of such a prognostic prediction model is illustrated well. However, I feel that this manuscript in its current format is lacking in some aspects: 

Major

1. The current cohort is comprised of 234 CD patients. However, only 14 present NDR against 220 that present DR. For the test set, the comparison is 4 NDR versus 0.2*220=44 DR (if I am not mistaken). Such an imbalance would be a challenge for any kind of machine learning application if not addressed properly. One way of mitigating the effect is by validating the observations in an independent cohort. 

2. Patients with NDR present a high degree of combination with immunosuppressant. Could that be causative?

3. While the concept of using genetics to impute transcriptomics is an interesting take, it does raise multiple questions. First and foremost, this approach appears at first hand to disregard the effects of other non-genetic factors despite the growing body of literature of, for example, epigenetics or transcriptional regulation in general on gene expression. A discussion thereof would be warranted within this context. In line with the previous, the most pertinent question that arises as a result thereof is whether the expression of the reported genes, such as DPY19L3 are actually differentially expressed. While I appreciate that the authors mention this in their discussion, I feel that the addition of gene expression data is a requirement to consider this manuscript for publication. 

Minor

1. The tissues that were used from GTEX should have been mentioned in the methods (as well) rather than in the results only.

2. During the quality control removal of the samples, what criteria were used for removing the 16 samples? Please provide the thresholds that were used for the call rate, the heterozygosity, relative relationship, and principal component analysis tests. Moreover, which software package (and version) was used to perform this QC? PLINK?

3. The authors only mention that they looked at anti-TNF, do they have more detail on which patients were on adalimumab and which ones on infliximab?

Other points

1. The authors have defined response based on a global physician assessment. This leaves a lot of leeway in how response is determined. This might have been a sufficient criterion if 100s of patients presented NDR or DR (assuming that the scores would become slightly more representable). Instead, more objective measures, such as fecal calprotectin, CRP. However, the authors properly addressed this in their discussion.

Author Response

<Reviewer 2>

In the current study, the authors investigated whether durable response (DR) to anti-TNF can be predicted from non-durable response (NDR) using gene expression as imputed from the genetics. Overall, the concept is novel and interesting and the clinical need of such a prognostic prediction model is illustrated well. However, I feel that this manuscript in its current format is lacking in some aspects: 

Major

1.The current cohort is comprised of 234 CD patients. However, only 14 present NDR against 220 that present DR. For the test set, the comparison is 4 NDR versus 0.2*220=44 DR (if I am not mistaken). Such an imbalance would be a challenge for any kind of machine learning application if not addressed properly. One way of mitigating the effect is by validating the observations in an independent cohort. 

Response: We admit that our dataset is very limited in size for the development of predictive models using machine learning. Unfortunately, we do not have an independent cohort to validate our result. As you indicated, we have only 4 NDR cases in the test set. In order to overcome the idiosyncrasy of a particular training-test split, we repeated the split 100 times and reported the distribution and average properties. As pointed out by the Reviewer 1, we now included precision-recall and PRC analyses. (Figure S1-S6, Table S1-S4) While our results retained a substantial amount of false positives as measured by precision and PRC, the PRC curves improved with more genes just like the ROC curves. If the primary goal of the clinical predictions such as ours were to screen potential NDR patients that should be closely followed up, we believed that these levels of false positives were acceptable.

  1. Patients with NDR present a high degree of combination with immunosuppressant. Could that be causative?

Response: Thank you for your important comments. There was error in our Table 1 that we misrepresented proportion of combination of immunosuppressants by changing the results between NDR and DR. Correct result was 10/14 (71.4%) in NDR and 206/220 (93.6%) in DR (P<0.001). Higher proportion of combination of immunosuppressants in DR group might affect durable response in DR group. However, unfortunately, we didn’t include other drug history in the clinical features for analysis. Thus we added the sentence in the limitation section of DISCUSSION as follows: “Third, except eight clinical variables, drug history such as combination of immunosuppressants were not considered in this analysis. Including more clinical variables, and compared with genetic features in the predictive model is needed.”  

  1. While the concept of using genetics to impute transcriptomics is an interesting take, it does raise multiple questions. First and foremost, this approach appears at first hand to disregard the effects of other non-genetic factors despite the growing body of literature of, for example, epigenetics or transcriptional regulation in general on gene expression. A discussion thereof would be warranted within this context. In line with the previous, the most pertinent question that arises as a result thereof is whether the expression of the reported genes, such as DPY19L3 are actually differentially expressed. While I appreciate that the authors mention this in their discussion, I feel that the addition of gene expression data is a requirement to consider this manuscript for publication. 

Response: We agree that gene expression regulation is a complex process. We by no means attempt to replace real gene expression values with those imputed from genotypes. Rather we try to summarize local genotypic information as the  expression of a nearby gene. We call this process the imputation of gene expression, which is defined by PrediXcan. We use only those genes whose imputation passed the filtering criteria set by PrediXcan. The imputed expressions are an inherited baseline property just like the genotypes. This baseline property can be desirable in developing predictive models, as it doesn't change over time or by environment. On the other hand, it doesn’t reflect other non-genetic factors, undermining the performance of the model. We tried to compensate for this with clinical features that were measured at diagnosis. Interestingly, the contribution from the clinical features was not large.

The reason we did not present real expression data of those marker genes in our cohort was that not all the blood samplings were prior to the initiation of anti-TNF therapy. Moreover, immunosuppressant medication was heterogeneous between the groups.  It is our future plan to collect true baseline whole blood of the cohort in order to measure the expression levels of the marker genes and evaluate their feasibility as the biomarkers of future NDR. In this regard, our description on post hoc eQTL analyses of these genes can be misleading. We revised sentence in the RESULT section (3.4. Genetic bases of selected features) of revised manuscript in such a way emphasizing concordant eQTL directions, not up/down-regulations.

 Minor

4.The tissues that were used from GTEX should have been mentioned in the methods (as well) rather than in the results only.

Response: Thank you for your valuable comments. We revised in the METHOD of revised manuscript as follows : “ Among the tissue models that PrediXcan v7 released, we considered that   gene models for the whole blood, transverse colon, and small intestine terminal ileum were relevant to CD, focusing the subsequent analyses to these tissues only. We calculated the imputed expression values of 6294, 5612, and 3107 genes for 234 patients in those three tissues, respectively. The imputed expression values were used as features in the predictive model development as described below. The selected genes were also tested for their associations with NDR/DR status using the univariate logistic regression implemented as a PrediXcan function.

  1. During the quality control removal of the samples, what criteria were used for removing the 16 samples? Please provide the thresholds that were used for the call rate, the heterozygosity, relative relationship, and principal component analysis tests. Moreover, which software package (and version) was used to perform this QC? PLINK?

Response: The inclusion thresholds were call rate > 0.95, heterozygosity within +/- 3 standard deviation of mean, proportion of IBD (pi_hat) < 0.20, within the main cluster of the principal coordinate analysis plot. All QC metrics were calculated with PLINK v1.90b6.4.  We added the sentence in the METHOD of revised manuscript as follows: “Sample quality control of call rate (> 95%), heterozygosity (within ±3 standard deviation of mean), relative relationship (proportion of IBD < 0.2), and principal component analysis (within the main cluster) tests, eliminated 16 samples, resulting in 878 samples. All QC metrics were calculated with PLINK (v1.90b6.4).”

  1. The authors only mention that they looked at anti-TNF, do they have more detail on which patients were on adalimumab and which ones on infliximab?

 Response: Thank you for your valuable comments. We included the patients who used first anti-TNF therapy consisting of infliximab or CT-P13, the biosimilar, and didn't include adalimumab. We described this in the METHOD section of original manuscript as follows:” Among the 894 patients, the inclusion criteria were as follows: 1) diagnosis of CD; and 2) first anti-TNF therapy consisting of infliximab or CT-P13”

Other points

  1. The authors have defined response based on a global physician assessment. This leaves a lot of leeway in how response is determined. This might have been a sufficient criterion if 100s of patients presented NDR or DR (assuming that the scores would become slightly more representable). Instead, more objective measures, such as fecal calprotectin, CRP. However, the authors properly addressed this in their discussion.

Response: Thank you for your valuable comments. We agree with your comments, thus we described related comments in our limitation section of original manuscript.

Round 2

Reviewer 1 Report

I have no further concerns.

Reviewer 2 Report

My comments have been largely addressed.